# Effect of Chitosan- and Alginate-Based Coatings Enriched with Cinnamon Essential Oil Microcapsules to Improve the Postharvest Quality of Mangoes

**DOI:** 10.3390/ma12132039

**Published:** 2019-06-26

**Authors:** Cheng Yin, Chongxing Huang, Jun Wang, Ying Liu, Peng Lu, Lijie Huang

**Affiliations:** 1Light Industry and Food Engineering College, Guangxi University, Nanning 530004, China; 2School of Mechanical Engineering, Jiangnan University, Wuxi 214122, China

**Keywords:** mango, cinnamon essential oil microcapsule, chitosan, sodium alginate, coating, preservation

## Abstract

During this study, fresh mangoes were packed into multilayer coatings made from chitosan containing cinnamon essential oil microcapsules and alginate solutions that were alternately deposited on the mango surfaces by electrostatic interaction. We then compared the physical and chemical indexes to examine the changes in the mangoes during 14 d of storage. The results showed that the microcapsules prepared in the experiment were of uniform size, with the sustained release of essential oil exceeding 168 h. Compared with uncoated mangoes, the mangoes coated with the coatings could effectively inhibit the decrease of the titratable acid, soluble solids, and vitamin C contents; slow down the increase of the weight loss and pH; delay the appearance of mango respiration peaks; and preserve the firmness at storage conditions of 25 °C and 50% RH. Our findings revealed that mangoes without treatment showed losses in their edible and commercial value after 14 days in storage, and the mangoes coated with five layers still retained food and commercial value. Cross-sectional scanning electron microscopy images of the coatings showed that they had distinct layers and were of good uniformity and tight binding, and they also had good adhesion to the mango surface. These findings provide important insights into the use of coatings for the packaging of fruits during storage, which is essential for promoting the application of coatings for packaging preservation without big cost and expensive equipment.

## 1. Introduction

Mangoes are rich in many nutritional elements, such as vitamins, micronutrients, phytochemicals, dietary fibre, and phenolic compounds [1]. However, mangoes ripen easily at room temperature after picking and are extremely sensitive to ethylene. At present, methods used for maintaining the freshness of mangoes include cold storage, atmosphere modification, radiation, cling film packaging and chemical antisepsis. Among these approaches, cold storage, atmospheric modification, radiation, and other methods require a large investment in equipment. Moreover, plastic wrap packaging made from polyvinyl chloride (PVC) and chemical antisepsis may pose threats to human health [2]. Coatings are natural, non-toxic, and harmless materials containing macromolecular polysaccharides, proteins, lipids, etc. The use of coatings has emerged as one of the major preservation methods used in recent years [3]. Coatings can protect food products from mechanical, physical, chemical, and microbial damage and can extend the shelf life [4]. Coatings can be applied to mangoes by dipping, painting, or spraying to form a thin, transparent coating on the mango surface [5].

Polysaccharide-based coatings can be used to control the internal atmosphere of fruits and retard the ripening process [6], because they provide a partial barrier to moisture, O_2_, and CO_2_, also preventing the loss of volatiles [7]. Chitosan is a cationic polysaccharide obtained by the deacetylation of chitin in an alkaline medium [8]. It has good antibacterial properties and selective permeability to ethylene, carbon dioxide, and oxygen [9], and it can inhibit the respiratory rates of fruits and vegetables, delay the loss of nutrients, reduce the loss in fruit quality, maintain fruit firmness, and retain freshness in fruits and vegetables [10,11]. The cationic properties of chitosan take advantage of electrostatic interactions with anions [12]. One of the major drawbacks to the application of chitosan films to fresh food products is their poor water vapour barrier properties [13], which can be improved through the incorporation of essential oils (EOs) [14].

EOs are efficient and safe natural food preservatives with natural antioxidants and antimicrobials. They have almost no side effects on human health, and their advantages are much greater than those of synthetic chemical preservatives [15]. Cinnamon essential oil (CEO) is usually a yellow or amber liquid with a specific sweet aroma [16]. It is resistant to a large number of bacteria, moulds, and yeasts. It is a highly efficient and environmentally friendly natural antioxidant, food stabilizer, and green preservative [17]. However, its use as a food preservative is limited due to its volatility and irritating nature [18]. Encapsulation technology can enhance the solubility, control the volatility, and mask the flavours and unpleasant odours of essential oils [19].

In addition, the adhesion of chitosan to the surfaces of fruit and vegetables is limited, and it may cause an uneven coating on their surfaces. Moreover, a single-layer coating film cannot meet practical needs. The development of multilayer coating films using the layer-by-layer (LbL) electrodeposition technique can overcome the shortcomings of single-layer coating films [20,21]. Sodium alginate is a natural polyanionic polysaccharide extracted from algae and fungi, which dissolves in water and is insoluble in organic solvents. It has good film formability and is non-toxic and biodegradable [22]. Furthermore, it can form a good atmosphere around the surfaces of fruits and vegetables, reduce the volatilization of moisture from fruits and vegetables, and restrain their respiratory rates [23]. Simultaneously, sodium alginate has good adhesiveness and gel properties, which causes ion exchange reactions with cationic materials to form sodium alginate gels or electrostatic layers of edible coatings to cover the surfaces of fruits and vegetables. Thus, this material can prevent the reproduction of microorganisms [24].

The existing methods for preserving fruits and vegetables are relatively expensive, and some of them are harmful to human health. Even if a preservation coating is environmentally safe, it only forms a single protective layer on the surfaces of fruits and vegetables, and the resulting effect is poor and lacks long-term effects. Therefore, the aim of this study is to combine microcapsule technology, LbL electrodeposition techniques, and coating technology to create multilayer coatings with multi-preservation environments on the surfaces of mangoes and to study the effects of different layers of coatings on the preservation effects of mangoes.

## 2. Materials and Methods

### 2.1. Preparation of Chitosan Inclusion Complex with Cinnamon Essential Oil

The method used for preparing the microcapsules was based on Feyzioglu’s two-step method for preparing chitosan-mint essential oil microcapsules [25]. A chitosan solution was prepared by dissolving 4.125 g of chitosan (90.0% deacetylated, Sinopharm Chemical Reagent Co., Ltd., Shanghai, China) in glacial acetic acid (1% v/v) (chemically pure, Tianjin Zhiyuan Chemical Reagent Co., Ltd., Tianjin, China), and the pH of this solution was adjusted to 4.8 using a 1 mol L^−1^ NaOH solution (analytically pure, Aladdin Reagent (Shanghai) Co., Ltd., Shanghai, China).

The surfactant Tween 80 (0.75% m/v) (chemically pure, Chengdu Kelon Chemical Reagent Co., Ltd., Sichuan, China) was added to the above chitosan solution and stirred by magnetic stirrer (ES35 A-Pro; Beijing Lab Tech Co., Ltd., China) for 30 min at 50 °C. After the temperature of the mixed solution was decreased to 25 °C, 1.375 g of cinnamon essential oil (90% purity, Zhuhai Weijia Food Additive Co., Ltd., Zhuhai, China) was added to the above mixed solution and stirred for 1 h to obtain a final oil-in-water emulsion.

A 0.5 g quantity of solid sodium tripolyphosphate (analytically pure, Aladdin Reagent Co., Ltd., Shanghai, China) was completely dissolved in distilled water and then added to the above emulsion and stirred for 30 min at a stirring rate of 500 r min^−1^ to form a milky white suspension. This suspension was centrifuged (Hema TGL-18R; Zhuhai Hema Medical Instrument Co., Ltd., Zhuhai, China) at 8832 G and 25 °C for 15 min. The lower layer was then removed, washed three times with ultrapure water, and shaken in a vortex mixer (XW-80A; Qilinbeier Instrument Manufacturing Co., Ltd., Haimen, China) for 15 s to obtain the final microcapsule suspension liquid. Finally, the microcapsule suspension was vacuum freeze-dried (DGL-10; Shanghai Qiaofeng Industrial Co., Ltd., Shanghai, China) for 48 h.

### 2.2. Coating Solutions and Preservation of Mangoes

#### 2.2.1. Chitosan Coating with Chitosan-Cinnamon Essential Oil Microcapsules

The chitosan coating was prepared as described by Elena et al. [26]. A 2.5 mL volume of glacial acetic acid was dissolved in 0.5 L distilled water and it was stirred and heated. A 1 g quantity of chitosan was dissolved in the above mixture when the temperature was 60 °C; stirring was then continued at a constant temperature of 60 °C for 1 h to obtain the chitosan solution. A 30 mL volume of the chitosan-cinnamon oil microcapsule suspension was then added to the prepared chitosan solution and stirred at 25 °C to obtain a chitosan solution containing microcapsules. Finally, the pH of the solution containing the microcapsules was adjusted to 3.0 using a 2 mol L^−1^ HCl solution (analytically pure, Lianjiang Ailianhua Reagent Co., Ltd., Zhanjiang, China).

#### 2.2.2. Alginate Coating

The alginate coating was prepared as described by Elena et al. [26]. The alginate coating solution was prepared by dissolving 1 g sodium alginate (200 ± 20 mPa·s viscosity, Aladdin Reagent Co., Ltd., Shanghai, China) in 0.5 L distilled water and stirring for 1 h to dissolve it completely to obtain a 0.2% (w/v) sodium alginate solution. The pH of this solution was adjusted to 7.0 using a 1 mol L^−1^ NaOH solution.

#### 2.2.3. Alginate-Chitosan Layer-by-Layer Coating

Mangoes (Baise, China), of the same size with no mechanical damage, and pests that were at the same ripening stage, were chosen. They were soaked in a 0.3% (v/v) sodium hypochlorite solution (40% purity, Aladdin Reagent Co., Ltd., Shanghai, China) for 5 min, rinsed three times using tap water, and dried in a sterile environment.

Following sterilization, the mangoes were randomly divided into four groups, with a control group and three test groups. Mangoes were treated as described by Elena et al. [26]. For the test groups, the mangoes were soaked in the sodium alginate solution (pH = 7.0) for 10 min. The residual electric charge on the mango surface was washed away using deionized water (pH = 7.0), and the mangoes were then dried at 25 °C for 20 min. This process formed the first layer of coating on the mango surface. Subsequently, the mangoes were soaked in the chitosan solution (pH = 3.0) containing microcapsules for 10 min, and the residual electric charge on the mango surface was washed off using deionized water (pH = 3.0). The mangoes were then dried at 25 °C for 20 min. This process added a second layer of coating to the mango surface using the electrostatic interaction between the sodium alginate solution and the chitosan solution containing microcapsules. The mangoes were then soaked in the sodium alginate solution again, and the above steps were repeated until three, five, and seven layers of coatings were assembled on the mango surface by alternate deposition. Finally, the mangoes in the control group (no layer) and the test groups were stored in an artificial climate chamber (CLIMACELL404; Germany MMM Company, Bavaria, Germany) at 25 °C and 50% relative humidity (RH), and the mango preservation indicators were tested and analysed at regular intervals.

### 2.3. Characterization of Microcapsules

#### 2.3.1. Surface Morphology of Microcapsules

The surface morphology of the microcapsules was observed by field-emission scanning electron microscopy (SU-8020; Hitachi Hi-tech Co., Ltd., Tokyo, Japan) at a voltage of 10 kV with using the method described by Karen et al. [27]. Microcapsules were prepared by vacuum freeze-drying the microcapsule suspension prepared in Section 2.1 for 48 h

#### 2.3.2. In Vitro Release of Chitosan-Cinnamon Essential Oil Microcapsules

This experiment was conducted by a modified method according to Hu et al. [28]. A 10 mL volume of a chitosan–cinnamon essential oil microcapsule suspension and 20 mL phosphate-buffered saline (PBS, pH = 7) were dispensed into a dialysis bag (molecular weight cut-off of 8000–14,000). The bag was placed in a conical flask, a 200 mL volume of PBS was added, and, finally, the conical flask was placed in an electric hot water bath (HHS-type; Shanghai Boxun Industrial Co., Ltd., Shanghai, China) at a constant temperature of 37 °C. At 2 h, 4 h, 8 h, 12 h, 24 h, 48 h, 72 h, 120 h, and 168 h. Then, 0.0002 L dialysate outside the dialysis tubing was accurately aspirated, and an equal volume of PBS buffer was added while continuing the shaking. The dialysate was solubilized and dissolved with ethyl ethanol (95% purity, Tianjin Zhiyuan Chemical Reagent Co., Ltd., Tianjin, China). The PBS was used as a blank reading. The absorbance was measured at 275 nm using a UV-visible spectrophotometer (SPECORD plus 50; Analytik Jena AG, Jena, Germany).

### 2.4. Characterization of the Coatings

#### 2.4.1. Zeta Potential Analysis of Sodium Alginate Solution, Chitosan Solution, and Chitosan Solution with Microcapsules

During this experiment, the zeta potential values of the alginate, chitosan, and chitosan solutions containing microcapsules were obtained by the modified method according to Kaiser et al. [29], where a granulometer (Nano ZSP; Malvern Instruments, Malvern, UK) was used to determine whether the target solution had the opposite charge. The samples were measured in triplicate.

#### 2.4.2. Surface Morphology of Coatings on the Mango Surface

The surface morphology of the coatings on the mango surfaces were observed by using the method described by Karen et al. [27], with a slight modification. The mango peels of the test groups were cut randomly using a stainless-steel blade and completely dried by vacuum freezing. They were then characterized by field-emission scanning electron microscopy at a voltage of 10 kV.

#### 2.4.3. Analysing the Contact Angles of the Mango Surface Coatings

The contact angle values of the surfaces were determined by KRUSS contact angle meter (DSA 100; Germany KRUSS Equipment Company, Hamburg, Germany), and the method was described by Veronesi et al. [30]. No layer represented the control mango surface; layers one, three, five and seven represented the sodium alginate solution coatings; and layers two, four, and six represented the chitosan solution coatings containing microcapsules. For each layer of the assembled coatings, three samples were used for testing, and the results were obtained by averaging the values.

### 2.5. Index Tests

#### 2.5.1. Weight Loss Rate

Using the direct weighing method described by Zhang et al. [31], the samples were weighed using an electronic balance (PL601-L; METTLER TOLEDO Instrument Co., Ltd., Zurich, Switzerland) before being coated. Before each sampling, the samples were weighed. Each operation was repeated three times, and the results were averaged. The weight loss rate was calculated as follows:(1)$$R_{w}\left(\%\right)=100\times \left(W_{i}-W_{0}\right)/W_{0}$$where R_W_ is the mango weight loss rate (%), W_i_ is the weight of the mango on day i, and W_0_ is the weight of the mango on day 0.

#### 2.5.2. pH

The pH values were measured according to the method of Zahedi et al. [32]. A 30 g mass of each mango’s pulp was homogenized in 150 mL of distilled water using a juicer (JYL-C020E; Nine Yang Co., Ltd., Laiwu, China) for 2 min and then filtered. The filtrate was collected and its pH was measured using the pH meter (FE28; Jinan Guangyao Medical Equipment Co., Ltd., Jinan, China) after the filtration. Each group of samples was measured three times, and the results were averaged.

#### 2.5.3. Firmness

In accordance with a previously method described by Jongsri et al. [33], the mangoes were peeled and a fruit firmness meter (GY-2 type; Shanghai Hu Yueming Scientific Instrument Co., Ltd., Shanghai, China) was used to measure the firmness of each fruit by penetrating the skin with the instrument to a depth of 1 cm at three different locations on the fruit (proximal, distal, and middle). Each group of samples was measured three times, and the results were averaged.

#### 2.5.4. Respiratory Rate

Using a previously method described by Gong et al. [34], the mango respiration rates were measured with the fruits and vegetable respiration apparatus (JFQ-315OH; Jun-Fang-Li-Hua Technology Institute, Beijing, China). The mangoes were individually placed in airtight jars and the valve was opened to allow air inside. Production was expressed as the CO_2_ concentration (mg CO_2_ kg^−1^ h^−1^). For each mango treatment, three mangoes were used per replication, and there were three replications.

#### 2.5.5. Titratable Acidity

As previously described by Yuan et al. [35], the sample solution was then titrated with a 0.1 M sodium hydroxide standard solution. Each group of samples was measured three times, and the results were averaged. The titratable acid content was calculated according to the following formula:(2)$$\mathrm{Acid}\;\mathrm{content}\;(\%)=\frac{V\times N\times K\times B}{b\times A}\times 100$$where V is the volume of sodium hydroxide solution consumed by the titration of the filtrate (mL), N is the NaOH solution concentration (M), K is the conversion factor (0.064) calculated using citric acid, B is the sample volume (mL), b is the filtrate volume used for titration (mL), and A is the sample weight (g).

#### 2.5.6. Soluble Solids

According to a previous method described by Jongsri et al. [33], a 25 g mass of peeled mango pulp was crushed and filtered through gauze. The filtrate was measured using a digital Abbe refractometer (WYA-2S; Shanghai Shen Guang Instrument Co., Ltd., Shanghai, China). All the samples were tested in triplicate.

#### 2.5.7. Vitamin C

In accordance with a previously described method [36], 25 g of the edible part of each sample was measured and mashed in a juicer. Next, 200 mL of oxalic acid solution (analytically pure, Tianjin Hengxing Chemical Preparation Co., Ltd., Tianjin, China) was added, and each sample was quickly homogenized. The above mixture was then added to a 200 mL volumetric flask and shaken. The filtrate plus kaolin decolourizer (analytically pure, Tianjin Kemiou Chemical Reagent Co., Ltd., Tianjin, China) were mixed and filtered. Next, 10 mL of the filtrate was titrated with a solution of 2,6-dichloroindophenol. Each group of samples was measured three times, and the results were averaged. The vitamin C content was calculated using the following formula:(3)$$\mathrm{Vitamin}\;C\;(g/\mathrm{kg})=\frac{\left(V-V_{0}\right)\cdot T\cdot A}{W}\times 100$$where V is the volume (mL) of the 2,6-dichloro-indophenol solution consumed by the titration sample, V_0_ is the volume (mL) of the 2,6-dichloro-indophenol solution consumed by the titration blank, T is the 2,6-dichloroindophenol solution titre (mg/mL), A is the dilution factor, and W is the sample weight (g).

#### 2.5.8. Chromatic Aberration

A spectrophotometer (CM-3600d; Japan Konica Minolta Company, Tokyo, Japan) was used to measure the colour of the mango pulp with the illuminant/viewing geometry of D65/10°. Each group was measured three times, and the results were averaged. Whiteboard and blackboard calibrations were performed before each measurement. The chromatic aberration was calculated as follows:(4)$$∆\mathrm{Eab}={\left(∆L^{2}+∆a^{2}+∆b^{2}\right)}^{1/2}$$where ∆E is the chromatism value of the mango pulp, L is the lightness index (L = 0 is used to represent black, L = 100 is used to represent white), a represents the red/green value (the value of +a is the red direction and the value of -a is the green direction), b represents the yellow/blue value (+b is the yellow direction and -b is the blue direction), L_0_ is the lightness on day 0, L_i_ is the lightness on day i, a_0_ is the red/green value on day 0, a_i_ is the red/green value on day i, b_0_ is the yellow/blue value on day 0, and b_i_ is the yellow/blue value on day 0.

### 2.6. Statistical Analysis

The data were analysed using an analysis of variance (ANOVA) with the SPSS 16.0 program (SPSS Inc., Chicago, IL, USA). Statistical correlations were evaluated using Pearson’s correlation coefficients, and a *p*-value < 0.05 was considered statistically significant. The figures were drawn with Origin 8 (OriginLab Corp., Northampton, MA, USA).

## 3. Results and Discussion

### 3.1. Surface Morphology of Microcapsules

As shown in Figure 1a, the size distribution of the chitosan–cinnamon essential oil microcapsules was approximately 200–300 nm, and the particle size distribution was uniform. In addition, the chitosan-cinnamon oil microcapsules had smooth and crack-free surfaces with uniform and regular spherical distribution, indicating that the cinnamon essential oil was well embedded in the microcapsules. As shown in Figure 1b, the size of the microcapsules without the cinnamon essential oil was obviously smaller than that of the chitosan-cinnamon essential oil microcapsules. We also concluded that the dispersion of microcapsules without cinnamon oil was not as good as that of the chitosan-cinnamon oil microcapsules, possibly due to the hydrophobicity of the cinnamon essential oil molecules embedded in the microcapsules [37].

### 3.2. In Vitro Release Experiment on Chitosan-Cinnamon Essential Oil Microcapsules

Figure 2 shows the sustained release of the chitosan–cinnamon essential oil microcapsules during in vitro release. As shown in the figure, within 2 h, the release rate of cinnamon essential oil from the chitosan-cinnamon essential oil microcapsules reached 48.5%. This rate occurred because the cinnamon essential oil was close to the surface of the microcapsules, thus resulting in easy release. Hence, at the beginning of the test, the cinnamon essential oil release rate was faster. After 12 h, the release rate slowed primarily because the chitosan wall material underwent swelling, dissolution, and desorption in PBS, which increased the porosity of the microcapsule surface and decreased the cinnamon essential oil that could be released from it [38]. The release rate of the cinnamon essential oil, which was released into the uniform release phase, was approximately 0.07% per hour. After 168 h, the release rate reached 94.2%, indicating that some of it was still not released.

In addition, the sustained release of the chitosan-cinnamon essential oil microcapsules in vitro indicates that the microcapsules will continue to work in coatings.

### 3.3. Zeta Potential Analysis

The self-assembly of each coating is driven by the electrostatic force generated by the change in charge on the coating surface [39]. The zeta potential analysis shows that when the pH of the chitosan solution is 3, the zeta potential value is +(64.1 ± 2.2) mV; when the pH of the sodium alginate solution is 7, the zeta potential value is −(66.0 ± 1.5) mV and the zeta potential value of the chitosan solution containing microcapsules is +(57.7 ± 3.6) mV. The above data indicate that the chitosan solution containing microcapsules with a positive charge and the sodium alginate solution with a negative charge can form a self-assembled coating on the mango surface by electrostatic interaction.

### 3.4. Morphological Distribution of Coatings on the Mango Surface

As shown in Figure 3a, the mango surface was clearly covered with the coating. Furthermore, it could be observed that the flatness of the coatings changed with a variation in the structure of the mango epidermis. This observation indicated that the coatings were tightly attached to the surface, and the formation and compactness of the coatings were very good. In addition, there were almost no holes on the surface. This finding also confirmed the good degree of binding by the coating to the mango surface.

Figure 3b shows the cross-section of coatings on the mango surface. We can intuitively observe that the coatings were made up of continuous layers. In combination, the thickness of the coatings in each layer showed better uniformity, distinct layers, and tight binding, indicating that the sodium alginate solution and the chitosan solution containing microcapsules could be coated using an electrostatic force.

### 3.5. Analysing the Contact Angle of the Mango Surface Coatings

To illustrate that the sodium alginate solution and the chitosan solution containing microcapsules could be alternately assembled on the mango surface through electrostatic interactions, we tested every layer (from no layer to seven layers) on the mango surface.

In general, higher contact angle values indicate that the surface of the material is more hydrophobic, whereas lower values are characteristic of hydrophilic surfaces [40]. In this test, for layers one, three, five, and seven (the sodium alginate layers), significantly smaller contact angles (46.64°–55.15°) were obtained, whereas for layers two, four, and six (containing microcapsules), high contact angle values (105.13°–108.04°) were observed. The evolution of the contact angle between each layer showed the alternate deposition of the sodium alginate solution and the chitosan solution containing microcapsules held together by electrostatic interactions. As shown in Figure 4, we also noted that the contact angle of the mango surface in the control group was 55.2 ± 0.3° (no layer), which represented hydrophilicity; simultaneously, the contact angle value for the first layer of the sodium alginate coating was 48.07 ± 0.8 °. The contact angle values of all the sodium alginate coating layers were similar, indicating that the sodium alginate solution had a lower surface energy value, it easily diffused on the mango surface, and it had a good adhesion ability. This finding also explained why the sodium alginate solution was chosen as the first coating in this study.

### 3.6. Changes in the Weight Loss Rate

When the mangoes are harvested, they continuously consume their own nutrients due to respiration and they continuously lose local water due to transpiration. These characteristics coupled with the absence of external water cause mangoes to wilt easily, resulting in tissue senescence and decreased freshness [41].

Figure 5a shows the effects from the different layers of coatings on the weight loss rate of mangoes during storage. As shown in Figure 5a, as the time of storage increased, the mangoes in both the control group and the test groups exhibited weight loss. From 0 d to 10 d, the weight loss rate of the mangoes coated with no layer and three, five, and seven layers was not significantly different (*p* > 0.05). At this time, the weight loss rate of the uncoated mangoes was 17.41%, and those of the mangoes coated with three, five, and seven layers were 17.12%, 15.93%, and 16.38%, respectively. On the 12th day, the uncoated mangoes showed a greater weight loss rate of 24.14%, while the mangoes coated with three, five, and seven layers showed a gradual linear increase. On the 14th day, the weight loss rate of the mangoes in the control group reached 30.97 %, while that of the mangoes coated with three, five, and seven layers reduced by 26.57%, 35.52%, and 30.58%, respectively. These changes indicated that the coating combined the high barrier properties of alginate and the antibacterial properties of chitosan, which could reduce the volatilization of the water vapour while reducing the O_2_ concentration on the mango surfaces and inhibiting the actions of respiratory enzymes, thereby slowing the respiratory rate of the mangoes. Finally, in a comparison of weight loss rates for the mangoes coated with three, five, and seven layers, it was found that the weight loss rate of the coated mangoes with five layers was the smallest (*p* > 0.05).

### 3.7. Changes in pH Value

As shown in Figure 5b, the pH of the mangoes in each group increased during storage. Among them, the pH value of the mangoes in the control group changed the most, followed by mangoes coated with three, five, and seven layers. The figure shows that during the first four days, the pH value in each group did not change significantly (*p* > 0.05). Subsequently, the pH value of the control group showed a clear upward trend. By the 14th day, the pH value of the control group reached a maximum value of 6.33. This change occurred because the organic acids that accumulated in the vacuoles of the mango pulp cells were gradually converted into sugars, and some of the remaining organic acids were consumed during respiration. Hence, the pH value of the cells remained high. For the test groups, the increasing trend in the pH value was significantly lower than that for the control group. This result occurred because the coating obstructed the external oxygen, inhibiting the respiration rate of the mangoes and reducing the loss of nutrients, and it slowed down the metabolism of the mangoes, reducing the rate of organic acid decomposition [42]. A similar increase in pH was reported in mango fruit stored for 8 d using a functional chitosan-lactoperoxidase system coatings [36].

### 3.8. Changes in Firmness

With increasing storage time, the pectin in mangoes gradually decomposes into soluble pectin, which decreases mango firmness. In addition, the hydrolysis and consumption of starch in mangoes promotes their softening. Therefore, firmness is a key indicator of mango freshness [43].

As shown in Figure 5c, the firmness of the mangoes in the control group decreased significantly faster than it did in the test groups after 14 d of storage (*p*
*<* 0.05). After the 10th day, the firmness of the mangoes in the control group decreased more steeply. Until the 14th day, the firmness value of the mangoes in the control group was 10.98 N, indicating that at that time, the uncoated mangoes were completely ripe, and their tissue texture had basically collapsed, making them almost inedible. At that time, the mangoes coated with three, five, and seven layers maintained good firmness levels of 56.55 N, 100.16 N, and 92.90 N, respectively, which indicated that the change in the softening rate depended on whether the mangoes were coated. After two weeks of storage, the mangoes coated with five and seven layers had the greatest effect on the mango texture. While the multilayer coatings showed good barrier properties, prevented the loss of moisture, and maintained the turgor pressure of the mango cells and their good firmness, the chitosan solution coating containing microcapsules inhibited bacterial growth and alleviated the degradation of the mango structure and tissues caused by microbial enzymes [44]. As shown in Figure 5c, we could also observe that the mangoes coated with seven layers had a slightly lower retention of firmness than those coated with five layers, which may be due to the increased deposition time of the mango in the soaking solution as the number of coating layers increased. The increasing concentration of the soaking solution leads to the infiltration and dehydration of the mango cells. As a result, the firmness of the mangoes coated with seven layers was lower than that of those coated with five layers. Alternatively, due to the increased deposition time, the coating solution parameters (such as the acidity of the chitosan solution) had some side effects on the mango’s epidermis. Therefore, the five-layer coating exhibited the best effect in terms of mango firmness. The changes in firmness of the mangoes with treatment were smaller than those of the previous study [36].

### 3.9. Changes in Respiration Rate

The respiration rate is an important indicator of fruit maturity. The greater the respiration rate, the faster the redox reaction rate in fruits and vegetables—that is, the faster the organic nutrients are consumed. Mangoes are a type of respiratory climacteric fruit, and their maturity tends to increase significantly after the peak of respiration. If this peak can be put off, the mango storage cycle can be prolonged [45].

As shown in Figure 5d, since the mangoes in the control group were in direct contact with the air, a respiratory peak intensity of 235.25 mg CO_2_ kg^−1^ h^−1^ appeared on the 4th day, then the respiratory rate dropped rapidly after four days of storage, indicating that the uncoated mango rapidly entered the postharvest and senescence stages. However, the mangoes coated with three, five, and seven layers all had a similar respiration rate after four days of storage, and this rate was significantly lower than that of the uncoated mangoes (*p* < 0.05). Moreover, the respiratory peaks of the coated mangoes were both put off to the 6th day. This finding showed that the coating on the mango surface formed a sealed space that regulated or prevented the exchange between the mango and the outside environment, creating a low-oxygen high-carbon dioxide environment in the membrane, which inhibited respiration, slowed down the metabolic intensity of the mangoes, and delayed the appearance of the mango’s respiration peaks [46]. The results were similar to the results obtained by Cissé et al., who showed that a chitosan-based coating could reduce the respiration rate of mangoes by decreasing O_2_ and increasing CO_2_ [36]. As shown in Figure 5d, we also observed that the coated mangoes slowly decreased their respiratory rate on the 6th day. At this time, the decreasing trend in the respiration rate of the mangoes in the control group was more obvious. This finding indicates that the coated mangoes reduced the intensity of ageing within their own tissue structure. The maturity of mangoes tends to increase significantly after the peak of respiration, as the nutrients of mangoes are then consumed. We also observed that after the storage of 6 d, the respiration rate of mangoes coated with five layers was relatively higher than that of the mangoes coated with three layers (*p* > 0.05). This indicates that mangoes coated with five layers are rich in nutrients, resulting in the higher respiration rate. By further comparison, the mangoes with five layers of coating showed the best respiration rate trend (better than that of the mangoes coated with seven layers (*p* > 0.05)), which may be due to the increased thickness of the coating in the latter, and this could lead to the anaerobic respiration of mangoes, which would also have a bad influence on the quality of the mangoes. Furthermore, considering that the cost of the coating with five layers is lower than that of the coating with seven layers, and the process of coating with five layers is simpler, the coating with five layers seems the most appropriate.

### 3.10. Changes in Titratable Acidity

The titratable acid content is one of the most important indicators in mangoes. In general, mangoes with a high sugar content and medium acidity are considered to be of superior quality. Titratable acid can help to maintain continuous mango respiratory activity, resist spoilage pathogens, and prevent mangoes from being attacked by microorganisms [47]. When mangoes are stored, with the participation of their enzyme systems, the titratable acid content is gradually converted into sugar nutrients, resulting in a continuous reduction in titratable acid content [48].

As shown in Figure 6a, the titratable acid content of the control and test groups showed a downward trend during storage. This result was caused by mango respiration, which resulted in the decomposition of the acid in the tissue cells. Organic acids were involved in many physiological reactions within the cell as intermediate metabolites, and they were consumed [49]. As indicated in Figure 6a, the organic acid content of the coated mangoes decreased at a slower rate because the coating reduced the respiration rate of the mangoes, slowed their metabolic reaction processes, and provided them with a fresh and full appearance. The acid content in the control group at 0–10 d showed a significant decline, until the titratable acid content was only 0.16% on the 10th day, indicating that the mango cells of the control group had weak metabolic activity and poor texture. On the 12th day, the titratable acid content of the mangoes in the control group increased again. This increase occurred because the mangoes had decayed and deteriorated, and the pathogens decomposed to produce acid, resulting in an increase in titratable acid content. On the 14th day of storage, the titratable acid content of the uncoated mangoes was 0.1%, while those of the mangoes coated with three, five, and seven layers were 0.29%, 0.38%, and 0.31%, respectively (*p* < 0.05). These changes indicated that the coatings could maintain a good titratable acid content. A similar reduction in titratable acidity was reported by a previous study [36]. After a further comparison of the titratable acid contents of the mangoes coated with three, five, and seven layers, it was found that those coated with five layers had the best effect in terms of inhibiting the reduction in the titratable acid content (*p* < 0.05).

### 3.11. Changes in Soluble Solid Content

Soluble solids are an important indicator of fruit and vegetable quality and are directly proportional to sugar content. The soluble solids of fruits and vegetables are also closely related to metabolism. Specifically, the stronger the respiration of fruits and vegetables is, the more substances are consumed, which reduces the soluble solid content of the fruits and vegetables. If the increasing trend in soluble solids were to be inhibited during storage, the shelf life of mangoes would be extended [50].

As shown in Figure 6b, throughout the storage period, the soluble solids of the mangoes in the control group first increased linearly (0–4 d) and then showed a slow growth trend (4–8 d). The soluble solids began to decline significantly during the later period (*p* > 0.05). This phenomenon could be explained as follows: during the storage period, the respiration rate of the mangoes increased continuously, and a large amount of organic matter was decomposed into sugars, acids, and minerals, resulting in an increase in the soluble solids. With an increase in storage time, the mangoes reached full maturity, and the soluble solids reached an extreme value. During the later period, due to the continuous progress of the respiratory process, coupled with the growth of spoilage bacteria and the decomposition of a large quantity of nutrients, the soluble solids decreased significantly, and the mangoes showed qualitative changes at this time [51]. In general, a coating with different layers effectively inhibited the increase in soluble solids during storage, which was similar to the results obtained by Khaliq et al. [52] From Figure 6b, we also observed that during the storage of the mangoes coated with three layers, the middle period term lasted longer, from the 4th day to the 12th day and then until the 14th day, when the soluble solids began to decline. The soluble solids of the mangoes coated with five and seven layers increased during the early stage of storage but gradually increased during the middle period and continued to increase until the 14th day. The two coatings significantly slowed down the rate of decline in the soluble solids and prolonged the late storage period of the mangoes. This was because the “low-oxygen high-carbon dioxide” environment created by the coatings reduced the metabolic activity of the mangoes, resulting in slower polysaccharide degradation and thus lowering the soluble solids [53].

### 3.12. Changes in Vitamin C

Vitamin C is considered as an important indicator for assessing damage in postharvest mangoes. The vitamin C in mangoes is very unstable because it is oxidatively decomposed easily, and its physiological activity diminishes during storage.

As shown in Figure 6c, the vitamin C content of the coated mangoes was significantly higher than that of the uncoated ones (*p* < 0.05). In particular, the vitamin C content of the mangoes coated with five and seven layers was relatively high after 14 d of storage (*p* < 0.05). This was because the coatings blocked oxygen from the external environment, inhibited the activity of ascorbates, and delayed the oxidation of vitamins. In addition, the cinnamon oil in the microcapsules of the chitosan coating possessed antioxidant properties, and the volatilization of the cinnamon oil inhibited the further oxidative breakdown of vitamin C, thereby maintaining a high vitamin C content while effectively delaying mango ripening [54]. The vitamin C content was also related to the degree of collapse in the tissue structure of the fruit. The destruction of this structure increased the density of the material, which affected the activity of the phenolic enzymes and the rate of vitamin C degradation [55]. This finding is consistent with previous results on the effects of coatings with different layers on mango firmness. That is, the loss of firmness was most rapid in the uncoated mangoes, resulting in the collapse of the cell wall structure and a decreased intercellular viscosity, which, in turn, promoted phenolic enzyme activity and increased the rate of vitamin C degradation.

### 3.13. Changes in Colour

Colour is an important indicator of the maturity of mangoes and the value of commodities. With increased storage time, chlorophyll is hydrolysed by enzymes to produce water-soluble substances, such as phytol and chlorophyll; coupled with photooxidation, the chlorophyll levels decrease or disappear, along with changes in the green colour [56].

Figure 7a shows the changes in the mango appearances after 14 days of storage at 25 °C and 50% RH. As shown in Figure 7a, after 14 days of storage, the peels of mangoes without coatings had turned yellow, were dehydrated and wrinkled, and many dark spots appeared on the surface of the peel after being eroded by mould, indicating a loss of commercial value due to spoilage. The peels of mangoes with three-layer coatings also turned yellow and displayed some black spots. The peels of mangoes with seven-layer coatings were greener than those of the previous two groups of mangoes. However, the mangoes with seven-layer coatings still had some dark spots on their surfaces. By contrast, mangoes coated with five layers had greener skin, a fresher colour, slight yellowing, and almost no visible dark spots, indicating that the coatings, to some extent, can slow down the yellowing of mangoes and prevent the formation of dark spots on their surfaces. Among the treatments, the mangoes coated with five layers retained their commercial and consumption value.

Figure 7b reflects the changes in the mango b values during storage. As the storage time increased, the b values of the mangoes increased, and the colour of the mangoes changed to yellow. As shown in Figure 7b, the b values of mangoes with no layer increased, and the colour of the mangoes gradually turned yellow, reaching their maximum value on day 12. These data indicated that the mangoes were completely mature. Before day 8, the b values of mangoes coated with three, five, and seven layers did not change substantially (*p* > 0.05). On day 14, the b value of mangoes coated with five layers, no layer, three layers, and seven layers were 36.49, 40.11, 36.68, and 37.45, respectively. This result indicated that the coating effectively delayed the time before the mango started yellowing (*p* > 0.05). This delay may have occurred because the five layers of coatings created an appropriately low O_2_ condition on the surfaces of the mangoes, inhibiting the oxidative decomposition of chlorophyll. The antibacterial properties of essential oil may have also inhibited the activity of chlorophyllase, resulting in fewer changes in the b value of the mangoes coated with five layers.

Figure 7c shows the colour changes in mangoes during storage. In general, the colour differences in mangoes tend to increase gradually during storage. The colour of the mangoes coated with no layer changed greatly during storage, and the colour difference value increased rapidly during storage (*p* > 0.05). On day 14, the colour difference value reached its maximum value. This is because the mangoes gradually matured as the storage time increased, and their flesh turned yellow. We also observed that the colour changes of coated mangoes were smaller than those of uncoated mangoes (*p* > 0.05). The slow release of essential oil from the coatings containing microcapsules resulted in reduced enzyme activity, protected the chlorophyll from hydrolysis and oxidation, and maintained the colour of the mangoes.

## 4. Conclusions

In this study, our results showed that untreated mangoes were perishable and suffered significant weight loss and colour changes. Moreover, the pH value clearly increased, whereas the firmness, respiration rate, vitamin C, soluble solid contents, and titratable acid contents decreased. Notably, these indicators were significantly improved in mangoes treated with coatings. Among the mangoes coated with three, five, and seven layers, coatings with five layers more effectively delayed mango decay and extended their shelf life, indicating obvious advantages in the preservation of mango nutrition and in maintaining various physical and chemical indicators in mangoes. After 14 days of storage, the untreated mangoes had lost their edible and commercial value, whereas mangoes coated with five layers still maintained their food and commercial value. Thus, our findings indicated that chitosan- and alginate-based coatings enriched with cinnamon essential oil microcapsules might be effective alternatives to improving the quality of mangoes. Furthermore, the coatings could have broad applications in the food industry. Furthermore, low molecular weight chitosan could perform at least equally to the coating, while showing improved physiological properties compared to higher molecular weight chitosan, suggesting hydrodynamic cavitation as an especially effective way to reduce molecular weight without altering chemical structure [57]. Thus, the selection and processing of raw materials for coatings should be carefully considered in future research.

## Figures and Tables

**Figure 1 materials-12-02039-f001:**
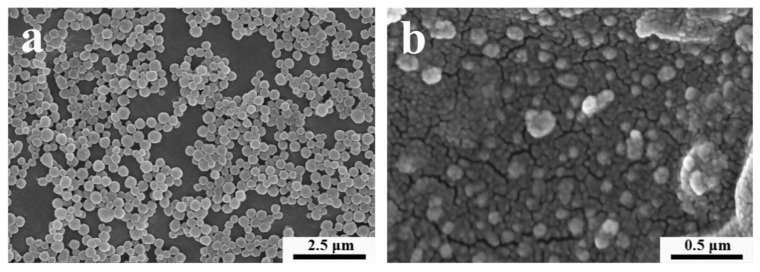
Field-emission scanning electron microscopy (FE-SEM) images of chitosan-cinnamon essential oil microcapsules (**a**) and chitosan particles without entrapped cinnamon oil (**b**) at 10,000× (**a**) and 100,000× (**b**) magnifications, respectively.

**Figure 2 materials-12-02039-f002:**
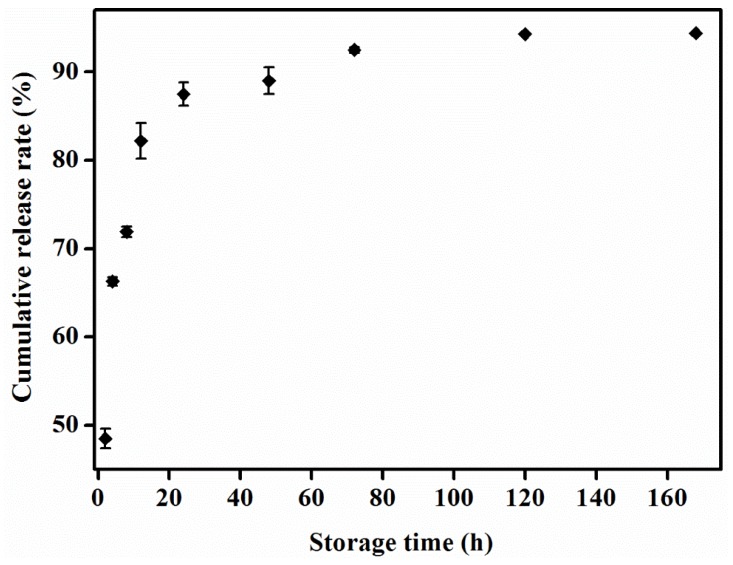
Cumulative release rate of chitosan–cinnamon essential oil microcapsules during in vitro release. Error bars represent the standard deviations of three replicates.

**Figure 3 materials-12-02039-f003:**
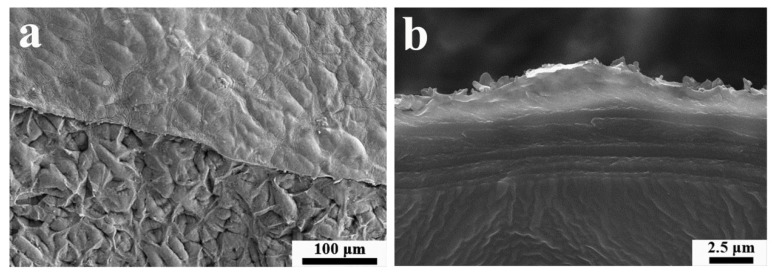
Field emission scanning electron microscopy (FE-SEM) images of multilayer coatings with seven layers on the mango surface (**a**) and a cross-section of the coatings with seven layers (**b**) at 250× (**a**) and 6000× (**b**) magnification, respectively.

**Figure 4 materials-12-02039-f004:**
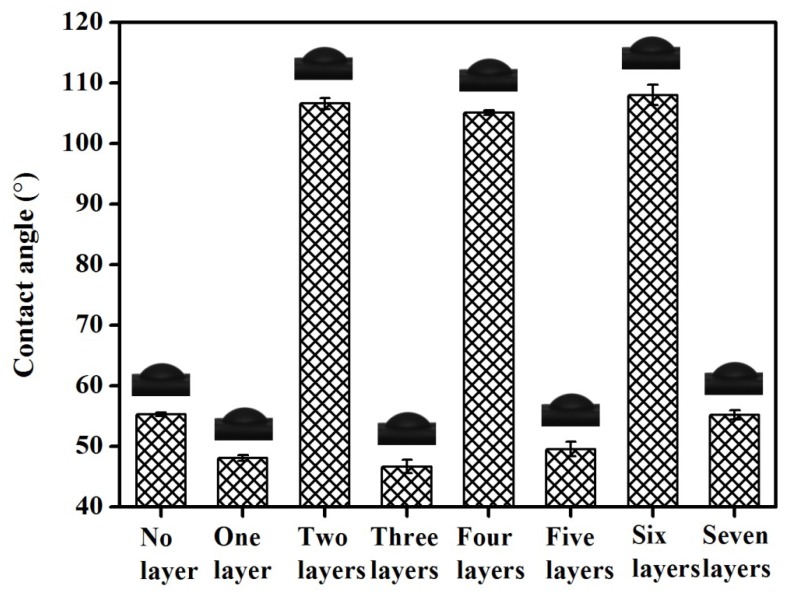
Contact angle values of different coating layers on the mango surface. Error bars represent the standard deviations of three replicates.

**Figure 5 materials-12-02039-f005:**
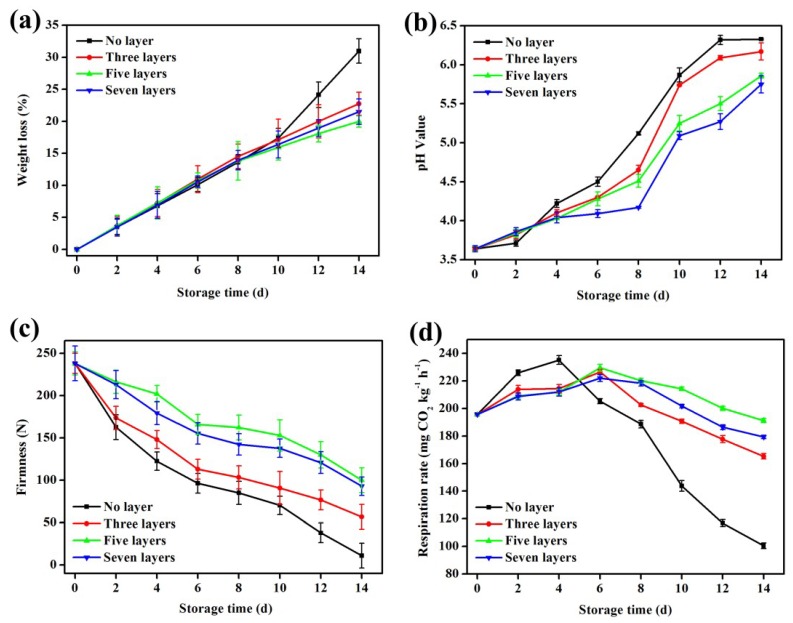
(**a**) Loss of quality during mango storage; (**b**) changes in the pH value during mango storage; (**c**) changes in firmness during mango storage; (**d**) changes in respiration rate during mango storage. Error bars represent the standard deviations of three replicates.

**Figure 6 materials-12-02039-f006:**
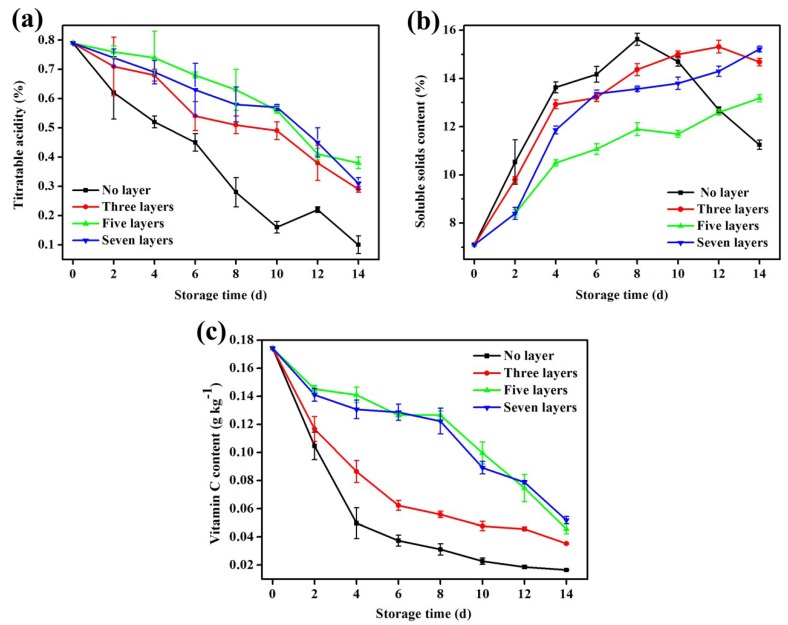
(**a**) Changes in the titratable acidity during mango storage; (**b**) changes in the soluble solid contents during mango storage; (**c**) changes in vitamin C during mango storage. The error bars represent the standard deviation of three replicates.

**Figure 7 materials-12-02039-f007:**
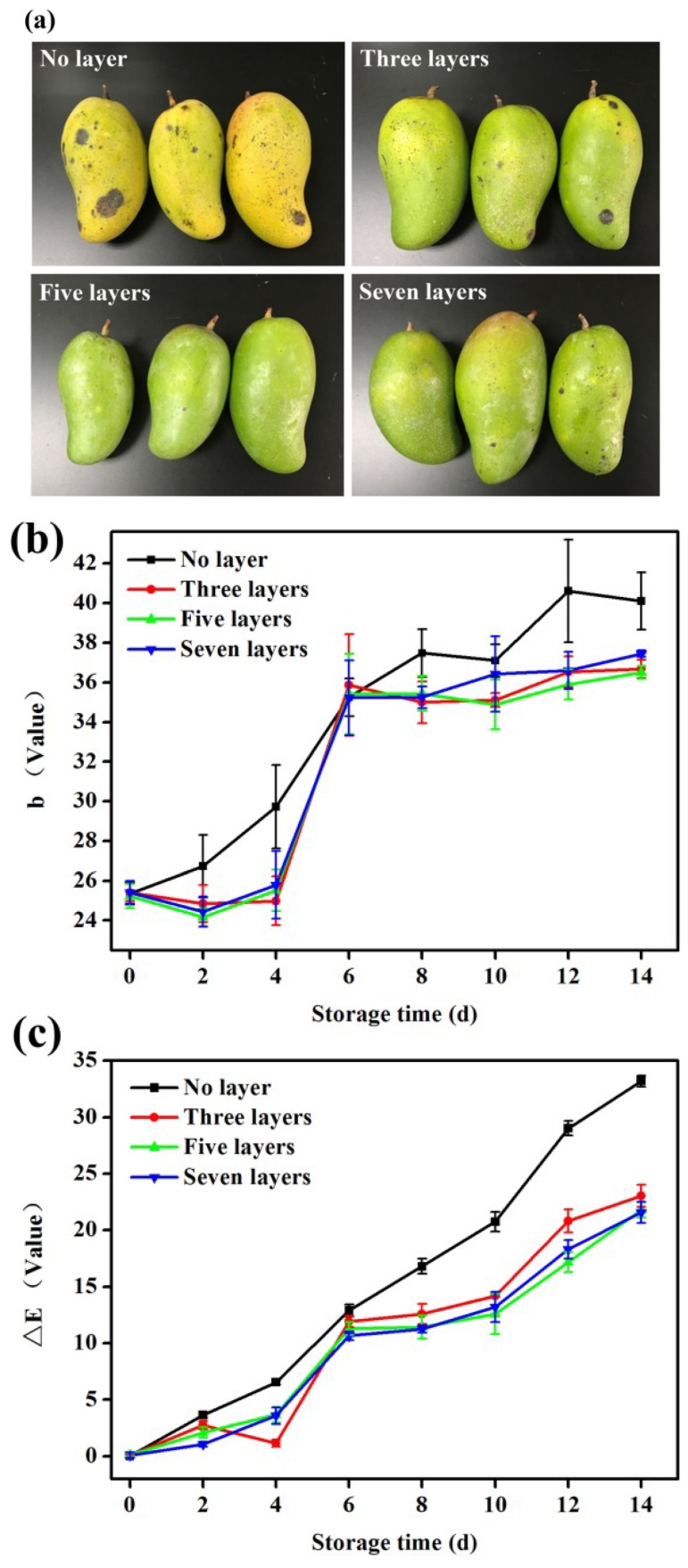
(**a**) Changes in the appearances of mangoes after 14 days of storage; (**b**) variations in chromaticity values (b value) during mango storage; (**c**) changes in colour during mango storage. Error bars represent the standard deviation of three replicates.

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
