# Peer review of "Effect of Chitosan- and Alginate-Based Coatings Enriched with Cinnamon Essential Oil Microcapsules to Improve the Postharvest Quality of Mangoes"

_materials, 2019, doi:10.3390/ma12132039_

Reviewer 1 Report

Edible film and coatings are one the emerging topics of todays' food science. Postharvest quality enhancement of mangoes presented by authors, was a novel and interesting approach. The research is designed appropriate with detailed presentation of methods and clear explanation of the results in a full extend research article. However, I have some general recommendation required prior to publication.

Point 1: In Materials and Methods section Reagents and Instruments are not usually presented in an individual section. It should be better to be incorporated in the following methods described.

Point 2: Sections 2.1, 2.4, 2.5, 2.6, 2.7.1, 2.7.2, 2.7.3, 2.7.5, 2.7.6, do not present any references of the methods used. If the described procedures are designed and experimented by authors it should be mentioned.

Point 3: The authors have not compared their results with other scientific data of similar studies i.e other use of coating in mangoes or other use of these coating in other fruits. They only present general explanation about observations.

Point 4: Highlights are usually presented below the abstract of the study

Author Response

Response to Reviewer 1 Comments

Paper entitled “Effect of chitosan- and alginate-based coatings enriched with cinnamon essential oil microcapsules to improve the postharvest quality of mangoes”. (Manuscript ID: materials-519826)

Dear Reviewer:

I would like to thank you for your constructive comments and suggestions to improve the quality of the paper.

As per your suggestion, we have revised the paper by:
Point 1: In Materials and Methods section Reagents and Instruments are not usually presented in an individual section. It should be better to be incorporated in the following methods described.

Response 1: Reagents and Instruments have been incorporated in the following methods described in the manuscript.

Point 2: Sections 2.1, 2.4, 2.5, 2.6, 2.7.1, 2.7.2, 2.7.3, 2.7.5, 2.7.6, do not present any references of the methods used. If the described procedures are designed and experimented by authors it should be mentioned.

Response 2: We have added references of the methods used in section 2.1, 2.4, 2.5, 2.6, 2.7.1, 2.7.2, 2.7.3, 2.7.5 and 2.7.6.

Point 3: The authors have not compared their results with other scientific data of similar studies i.e other use of coating in mangoes or other use of these coating in other fruits. They only present general explanation about observations.

Response 3: We have compared our results of pH, firmness, respiration rate, titratable acidity and soluble solids content with other scientific data of similar studies and have added the references.

Point 4: Highlights are usually presented below the abstract of the study.

Response 4: Highlights have been presented below the abstract of the study.

Reviewer 2 Report

with cinnamon essential oil microcapsules to improve the postharvest quality of mangoes.

Abstract

This section is vague. Please add your main results.

Materials and methods

Line 145- “Surface morphology of microcapsules”, sample preparation???

Line 151- “The bag was placed in a conical flask, some PBS was added, and finally, the conical flask was placed in an electric hot water bath.”?? Temperature???

Line 220- “Chromatic aberration”, illuminant and ºobserver used???

Results and discussion

Line 354- “As shown in Figure 5c, the firmness of the mangoes in the control group decreased significantly faster than it did in the test groups after 14 d of storage (P>0.05).”??? “(P>0.05)”?? or if is related to control group is “(P<0.05)”??< span="">

References

Please format scientific names in italic.

Author Response

Response to Reviewer 2 Comments

Paper entitled “Effect of chitosan- and alginate-based coatings enriched with cinnamon essential oil microcapsules to improve the postharvest quality of mangoes”. (Manuscript ID: materials-519826)

Dear Reviewer:

I would like to thank you for your constructive comments and suggestions to improve the quality of the paper.
As per your suggestion, we have revised the paper by:

Abstract

Point 1: This section is vague. Please add your main results.

Response 1: We have added our main results in abstract.

Materials and methods

Point 2: Line 145- “Surface morphology of microcapsules”, sample preparation???

Response 2: We have added the details of sample preparation of microcapsules in this part.

Point 3: Line 151- “The bag was placed in a conical flask, some PBS was added, and finally, the conical flask was placed in an electric hot water bath.”?? Temperature???

Response 3: The temperature was a constant temperature of 37 °C and it was added to the manuscript.

Point 4: Line 220- “Chromatic aberration”, illuminant and ºobserver used???

Response 4: The illuminant and viewing geometry were D65 and 10º, respectively.

Results and discussion

Point 5: Line 354- “As shown in Figure 5c, the firmness of the mangoes in the control group decreased significantly faster than it did in the test groups after 14 d of storage (P>0.05).”??? “(P>0.05)”?? or if is related to control group is “(P<0.05)”??< span="">

Response 5: After checking the statistical correlations, we have corrected the (P>0.05) to (P<0.05).< span="">

References

Point 6: Please format scientific names in italic.

Response 6: We have formatted scientific names in italic in references.

Reviewer 3 Report

General comments

The article deals with an experimental study about shelf life extension of mangoes by means of multi-layer alternate coatings based on natural materials - chitosan with cinnamon essential oil microcapsules, and alginate.

The study is well designed, sufficiently original although based on extensive previous knowledge, very well written and organized, and conveys clear results and perspective for the very important scientific and industrial field of post-harvest management of vegetable food, expecially fruit.

General comment No. 1An important missing point concerns the expected cost (affordability) of the proposed technical solution. The Authors should make an effort to assess the industrial-scale cost, for example in weight unit (e.g., per 100 kg or so), as compared to the market value of the coated product, with reference to the best option - five layers coating. Otherwise, the practical value of this research remains unknown.

General comment No. 2 - A very recent and comprehensive review could be cited and shortly discussed in the Introduction, as it deals, among the other, with the application of chitosan and cinnamon extracts to the postharvest preservation of citrus fruits (doi:10.3390/plants8020026).

General comment No. 3 - Some Authors found that low molecular weight chitosan could perform at least equally as a coating, while showing improved physiological properties compared to higher molecular weight chitosan, proposing hydrodynamic cavitation as an effective and especially effective means to reduce the molecular weight without altering its chemical structure. For example (doi:10.1016/j.polymdegradstab.2012.11.001) and references therein (and possibly other articles citing the above-mentioned one). A short discussion could be performed in the Conclusions, along with proper recommendations for further research.

Other specific comments

Line 31. "threats to human health". Explain how cling film can pose threats.

Lines 31-32. "These coatings...". What are "these coatings"? Are they the coatings considered in this study? Anyway, the statement is unclear and needs rephrasing.

Line 134. "the mangoes was". Change to "the mangoes were".

Lines 210-211. "Next, 200 mL of extractant was added, and each sample was quickly pounded into a homogenate". What are the extractant and the homogenate?

Line 217. "where V is the titration sample...". Likely, this is V − V0.

Lines 290-291. This statement is speculative and unsupported. The Authors should supply more arguments and reference(s) to to support the claim.

Figure 5 should be moved to subsection 3.6, where it is first referred to.

Line 330. "26.57 %, 35.52 %, and 30.58 %,". Looking at Figure 5a, these levels look like to be substantially smaller. Please check.

Line 335. "it was found that... was the smallest". The difference between 5 and 7 layers does not seem to be significant. Please check.

Lines 397-399. "The breathability of the mangoes... not as good". Not so convincing. If the overall breathability is lower, should not preservation be better? I'm not contesting this conclusion, but it should be better argumented. Why 5 layers are better than 3 or 7 layers?

Line 400. "3.10. Changes in titratable acidity". The related discussion could be merged with that in subsection 3.7 about the pH, due to the strict relation between titrable acidity and pH.

Lines 493-494. "the b value of mangoes... and seven layers". Actually, the differences seem very small, especially between five and three layers. Please check.

Lines 505-506. "the colour difference... was minimal". Difference between five and seven layers appear minimal and not significant. Please check.

Line 521. "coatings with five layers effectively delayed". Change to "coatings with five layers more effectively delayed". This, because often differences were quite tiny.

Author Response

Response to Reviewer 3 Comments

Paper entitled “Effect of chitosan- and alginate-based coatings enriched with cinnamon essential oil microcapsules to improve the postharvest quality of mangoes”. (Manuscript ID: materials-519826)

Dear Reviewer:

I would like to thank you for your constructive comments and suggestions to improve the quality of the paper.
As per your suggestion, we have revised the paper by:

General comments

Point 1: General comment No. 1 - An important missing point concerns the expected cost (affordability) of the proposed technical solution. The Authors should make an effort to assess the industrial-scale cost, for example in weight unit (e.g., per 100 kg or so), as compared to the market value of the coated product, with reference to the best option - five layers coating. Otherwise, the practical value of this research remains unknown.

Response 1: We calculated the mass of solid reagents used in the study. The results showed that a 8 g quantity of solid reagents could treat a 24 kg quantity of mangoes and the solid reagents we used in the study were very common and not expensive. What’s more, after the 14 d of storage, we can observed from figure 7a that black spots appeared on the surface of the mangoes coated with no layer, three layers and seven layers, while there was almost no black spot appearing on the surface of the mangoes coated with five layers. It indicated that the mangoes coated with five layers will be more likely to be purchased and it will produce good economic benefits.

Point 2: General comment No. 2 - A very recent and comprehensive review could be cited and shortly discussed in the Introduction, as it deals, among the other, with the application of chitosan and cinnamon extracts to the postharvest preservation of citrus fruits (doi:10.3390/plants8020026).

Response 2: According to the reviewer’s suggestion, paper entitled “Inhibition of Key Citrus Postharvest Fungal Strains by Plant Extracts In Vitro and In Vivo: A Review” (doi:10.3390/plants8020026) was cited and shortly discussed in Line 48-49 in the revised manuscript.

Point 3: General comment No. 3 - Some Authors found that low molecular weight chitosan could perform at least equally as a coating, while showing improved physiological properties compared to higher molecular weight chitosan, proposing hydrodynamic cavitation as an effective and especially effective means to reduce the molecular weight without altering its chemical structure. For example (doi:10.1016/j.polymdegradstab.2012.11.001) and references therein (and possibly other articles citing the above-mentioned one). A short discussion could be performed in the Conclusions, along with proper recommendations for further research.

Response 3: According to the reviewer’s suggestion, paper entitled “Degradation of chitosan by hydrodynamic cavitation” (doi:10.1016/j.polymdegradstab.2012.11.001) was cited and a short discussion was performed in Line 598-603 of Conclusions in the revised manuscript.

Other specific comments

Point 4: Line 31. "threats to human health". Explain how cling film can pose threats.

Response 4: “cling film” in the manuscript has been changed to “ plastic wrap” and the threats of plastic wrap to human health have been shortly discussed and a paper was cited in Line 46-47 in the revised manuscript.

Point 5: Lines 31-32. "These coatings...". What are "these coatings"? Are they the coatings considered in this study? Anyway, the statement is unclear and needs rephrasing.

Response 5: The statement submitted by reviewer has been rephrased in Line 47-48 in the revised manuscript.

Point 6: Line 134. "the mangoes was". Change to "the mangoes were".

Response 6: "the mangoes was" has been changed to "the mangoes were".

Point 7: Lines 210-211. "Next, 200 mL of extractant was added, and each sample was quickly pounded into a homogenate". What are the extractant and the homogenate?

Response 7: The extractant was oxalic acid solution and the homogenate that we wanted to express was the mixture obtained by homogenating. We have rephrased the expression in Line 256-258 in the revised manuscript.

Point 8: Line 217. "where V is the titration sample...". Likely, this is V − V0.

Response 8: We have already added a description to the letters of V and V0 in the formula in Line 246-266 in the revised manuscript.

Point 9: Lines 290-291. This statement is speculative and unsupported. The Authors should supply more arguments and reference(s) to to support the claim.

Response 9: After making a discussion of the statement, we have deleted the statement.

Point 10: Figure 5 should be moved to subsection 3.6, where it is first referred to.

Response 10: According to the reviewer’s suggestion, figure 5 has been moved to subsection 3.6.

Point 11: Line 330. "26.57 %, 35.52 %, and 30.58 %,". Looking at Figure 5a, these levels look like to be substantially smaller. Please check.

Response 11: We checked the results and these values are a percentage reduction from the value of the mangoes without treatment. It did not indicate a specific value for direct decline. We just want to make people more intuitive to see the clear differences of weight loss rate between coated mangoes and uncoated mangoes.

Point 12: Line 335. "it was found that... was the smallest". The difference between 5 and 7 layers does not seem to be significant. Please check.

Response 12: We checked the values of weight loss rate of mangoes coated with five and seven layers and the results showed that the values of them were19.97%±0.9% and 21.5%±2%, respectively. And we added the statistical correlation in Line 384 in the revised manuscript.

Point 13: Lines 397-399. "The breathability of the mangoes... not as good". Not so convincing. If the overall breathability is lower, should not preservation be better? I'm not contesting this conclusion, but it should be better argumented. Why 5 layers are better than 3 or 7 layers?

Response 13: We carefully re-discussed the results and we have reinterpreted the advantages of five layers in Line 451-456, 458-462 and added reference in Line 446-448 to better illustrate the advantages  of five layers than three layers and seven layers in the revised manuscript.

Point 14: Line 400. "3.10. Changes in titratable acidity". The related discussion could be merged with that in subsection 3.7 about the pH, due to the strict relation between titrable acidity and pH.

Response 14: Actually there is a strict relation between pH and titratable acidity. The pH values we detected were used to reflect the acid-base state of mangoes while the titratable acidity was used to detect the content of organic acids. The methods of detecting the two indexes are different and we want to show more indexes of mangoes with treatment and without treatment so that we can perform a better analysis of the coating. Thus, we want to remain these two parts and make analysis of them separately.

Point 15: Lines 493-494. "the b value of mangoes... and seven layers". Actually, the differences seem very small, especially between five and three layers. Please check.

Response 15: We have added the specific values of the b value of coated and uncoated mangoes and we replaced the previous statement. And we focused on explaining the advantages of coating. (Line 562-564)

Point 16: Lines 505-506. "the colour difference... was minimal". Difference between five and seven layers appear minimal and not significant. Please check.

Response 16: We check the statement and we thought that the previous statement was not very suitable. So we rephrased the statement and we focused on describing the colour changes of coated mangoes were smaller than that of uncoated mangoes. What’s more , we have added the statistical correlation. (Line 574-577)

Point 17: Line 521. "coatings with five layers effectively delayed". Change to "coatings with five layers more effectively delayed". This, because often differences were quite tiny.

Response 17: According to the reviewer’s suggestion, we have changed the statement to  "coatings with five layers more effectively delayed".

Reviewer 4 Report

The article entitledEffect of chitosan- and alginate-based coatings enriched with cinnamon essential oil microcapsules to improve the postharvest quality of mangoes " has been carefully reviewed. This is an interesting attempt towards the use of multilayer coatings with an essential oil microcapsules to improve quality of mangoesUnfortunately, adequate methods for assessing and achieving the goal have not been selected properly.  

And the “quality of foods may be defined as the composite of those characteristics that differentiate individual units of a product, and have significance in determining the degree of acceptability of that unit to the user” Kramer stated in 1965. It means that very important aspect is also food safety and its microbial state. So the lack of antimicrobial effect in presented experiment  in the paper is its disadvantage.  

I do not completely understand why the release of chitosan-cinnamon essential oil microcapsules was performed with PBS. You were checking the effect on moango and you should choose other technic, such as gas chromatography–mass spectrometry and gas chromatograpgy to evaluate the release kinetics of the oil from the coating. 

What would happen if you did not wash off mango electric charge after soaking? 

In my opinion the experiment is very complecated to perform. There are so many steps to achive good result that my question is : Is there any other prevention techniqe which may give similar results? MAP? VA?

Author Response

Response to Reviewer 4 Comments

Paper entitled “Effect of chitosan- and alginate-based coatings enriched with cinnamon essential oil microcapsules to improve the postharvest quality of mangoes”. (Manuscript ID: materials-519826)

Dear Reviewer:

I would like to thank you for your constructive comments and suggestions to improve the quality of the paper.
As per your suggestion, we have revised the paper by:

Point 1: The article entitled “Effect of chitosan- and alginate based coatings enriched with 

cinnamon essentialoil microcapsules to improve the postharvest quality of mangoes "has 

been carefully reviewed. This is an interesting attempt towards the use of multilayer 

coatings with an essential oil microcapsules to improve quality of mangoes. Unfortunately, 

adequate methods for assessing and achieving the goal have not been selected properly.

Response 1: The purpose of the study was to preserve postharvest mangoes with safe, green and simple methods. During the study, we first prepared the cinnamon essential oil microcapsules , and then we prepared chitosan solution containing cinnamon essential oil microcapsules and alginate solution, respectively. Next, the above solutions were alternately deposited on the mango surfaces by electrostatic interaction. We then compared the physical and chemical indexes to examine the changes in the mangoes during 14 d of storage. The results showed that the microcapsules prepared in the experiment were of uniform size, with the sustained release of essential oil exceeded 168 h. Compared with uncoated mangoes, the mangoes coated with the coatings could effectively inhibit the decrease of the titratable acid, soluble solids, and vitamin C contents; slow down the increase of the weight loss and pH; delay the appearance of mango respiration peaks and preserve the firmness at storage conditions of 25 °C and 50 % RH. Our findings revealed that mangoes without treatment showed losses in their edible and commercial value after 14 days in storage, and the mangoes coated with five layers did not still retained food and commercial value. Cross-sectional scanning electron microscopy images of the coatings showed that they had distinct layers and are of good uniformity and tight binding, and they also had good adhesion to the mango surface. These findings provided important insights into the use of coatings for the packaging of fruits during storage, which is essential for promoting the application of coatings for packaging preservation without big cost and expensive equipment.

Point 2: And the “quality of foods may be defined as the composite of those characteristics 

that differentiate individual units of a product, and have significance in determining the 

degree of acceptability of thatunit to the user” Kramer stated in 1965. It means that very

important aspect is also food safety and its microbial state. So the lack of antimicrobial 

effect in presented experiment in the paper is its disadvantage. 

Response 2: In the choice of raw materials, we chose edible essential oils and edible polysaccharides to prepare the coating to preserve the mangoes. And we focused on analysing the physical and chemical indexes of mangoes with treatment and without treatment. The results showed that the coating could effectively keep nutrient indicators of mangoes at a good nutritional level. After considering the preserving effect, treatment process and cost, we believe that five layers of coating could better preserve mangoes. What’s more, the black spots on the surface of mangoes are closely related to their infected microorganisms. We can observed from figure 7a that black spots appeared on the surface of the mangoes coated with no layer, three layers and seven layers, while there was almost no black spot appearing on the surface of the mangoes coated with five layers. It could indirectly prove the antibacterial effect of the coating.

Point 3: I do not completely understand why the release of chitosan cinnamon essential oil 

microcapsules was performed with PBS. You were checking the effect on mango and you

should choose othertechnic, such as gas chromatography–mass spectrometry and gas chromatog-rapgy to evaluate the release kinetics of the oil from the coating.

Response 3: We placed our samples together in an artificial climate chamber at storage conditions of 25 °C and 50 % RH to create a simulated real environment instead of packing mangoes in a separate package. Therefore, we believe that it is not our focus to preserve the mangoes by volatilizing the essential oil in the microcapsules. We believe that it is more important to achieve the preservation by directly contacting the essential oil in the microcapsules with the surface of the mango. So the release of microcapsules was performed with PBS and we have added the reference in this section.

Point 4: What would happen if you did not wash off mango electric charge after soaking?

Response 4: The purpose of using deionized water to clean mango surface is to wash off the electric charge on the surface of the mangoes. If we don’t take this process, the electric charge on the surface of mango is unevenly distributed when mango is picked up in solution. And we could wash off excess charge on the surface of mango to ensure the chitosan and alginate solutions can be alternately deposited on the mango surfaces by electrostatic interaction.

Point 5: In my opinion the experiment is very complecated to perform. There are so many 

steps to achieve good result that my question is : Is there any other prevention technique 

which may give similar results? MAP? VA? 

Response 5:

MAP has high requirements on the performance of packaging materials. For example, the breathability and moisture resistance of the packaging materials need to meet certain requirements, and the packaging materials also need to have a certain bearing capacity for fruits and vegetables. What’s more, MAP will affect the transportation capacity due to the protective gas entering the packaging environment and make the package bigger. The costs of protective gas and packaging materials are relatively high.

VA also has high requirements on the performance of packaging materials. For example, the breathability and moisture resistance of the packaging materials need to meet certain requirements, and the packaging materials also need to withstand the pressure when vacuuming. What’s more, VA will lead to anaerobic respiration of mangoes to some extent, which will also have a bad influence on the quality of mangoes.

In our study, we calculated the mass of solid reagents used in the study. The results showed that a 8 g quantity of solid reagents could treat a 24 kg quantity of mangoes and the solid reagents we used in the study were very common and not expensive. What’s more, when we carry out the treatment process, we only need to prepare the solutions in advance, and then impregnate the mangoes in order, which is easy to operate.

Round  2

Reviewer 3 Report

The Authors successfully managed to revise the paper, based on my coments and the comments provided by the other esteemed Reviewers.

Both corrections made to the manuscript, and responses - even critical ones - are appreciable and functional to a better result.

Now, the manuscript can be published. However, few language errors have been introduced with the newly introduced text, but these are quite minor errors that can be easily fixed at the proofs stage.